# Urinary urgency acts as a source of divided attention leading to changes in gait in older adults with overactive bladder

**William Gibson**[1]*, **Allyson Jones**[2], **Kathleen Hunter**[3], **Adrian Wagg**[1]

**1** Division of Geriatric Medicine, Faculty of Medicine and Dentistry, University of Alberta, Edmonton, Alberta, Canada, **2** Department of Physical Therapy, Faculty of Rehabilitation Medicine, University of Alberta, Edmonton, Alberta, Canada, **3** Faculty of Nursing, University of Alberta, Edmonton, Alberta, Canada

* wgibson@ualberta.ca

**Data Availability Statement:** The raw data are available at https://doi.org/10.7939/DVN/HXJYSA.

## Abstract

### Aims

There is a well-recognised but unexplained association between lower urinary tract symptoms including urgency and urgency incontinence and falls in older people. It has been hypothesised that urinary urgency acts as a source of divided attention, leading to gait changes which increase falls risk. This study aimed to assess whether urinary urgency acts as a source of divided attention in older adults with overactive bladder (OAB).

### Methods

27 community-dwelling adults aged 65 years and over with a clinical diagnosis of OAB underwent 3-Dimensional Instrumented Gait Analysis under three conditions; bladder empty, when experiencing urgency, and when being distracted by the n-back test. Temporal-spatial gait and kinematic gait data were compared between each condition using repeated measures ANOVA.

### Results

Gait velocity decreased from $1.1ms^{-1}$ in the bladder empty condition to $1.0ms^{-1}$ with urgency and $0.9ms^{-1}$ with distraction ($p = 0.008$ and $p<0.001$ respectively). Stride length also decreased, from 1.2m to 1.1m with urgency and 1.0m with distraction ($p<0.001$ for both). The presence of detrusor overactivity did not influence these results ($p = 0.77$).

### Conclusions

In older adults with OAB, urinary urgency induced similar changes in gait to those caused by a distracting task. These gait changes are associated with increased fall risk. This may be part of the explanation for the association between falls and lower urinary tract symptoms in older people. Future research should examine the effect of pharmacological treatment of OAB on gait and on the effect of dual-task training on gait when experiencing urgency.

**Funding:** This research was generously funded by the Canadian Urological Association/Astellas Research Grant.

**Competing interests:** The authors have declared that no competing interests exist.

## Introduction

Falls are the sixth leading cause of death in older adults, with the deaths of 2,691 Canadian seniors attributed to falls in 2008 [1]. Up to one third of people aged over 65, and half of those over 80, will fall in any given year [2]. Falls are often recurrent, with around half of people who fall experiencing another within 12 months [3]. Falls impair quality of life, cause individual pain and suffering, lead to functional decline [4], cause a fear of further falls [5] and are a significant cause of health resource use [6–8].

Urinary incontinence (UI), the involuntary leakage of urine, and lower urinary tract symptoms (LUTS) including urgency, the sudden compelling desire to void which is difficult to defer [9] are common [10, 11]. The prevalence, particularly that of urgency and urgency incontinence, rises in association with increasing age [12], with 7.1% of men and 9.7% of women aged under 40 reporting urgency compared to 19.1% of men and 18.3% of women aged over 60 [10]. The increasing prevalence of LUTS with age is likely to be due to multiple factors including age-related changes to the lower urinary tract and central nervous system, and increasing prevalence of concurrent medical conditions and polypharmacy [13]. The most common cause of urinary incontinence in older adults is overactive bladder (OAB), the clinical syndrome of urinary urgency, usually accompanied by increased daytime frequency and/or nocturia, with urinary incontinence (OAB-wet) or without (OAB-dry), in the absence of urinary tract infection or other detectable disease [14].

There is a strong but unexplained association between LUTS and falls in older adults. In community-dwelling older women, those with at least weekly urgency incontinence had a higher rate of falls than those without, with an age-adjusted odds ratio (OR) of 1.46 (95%CI 1.32–1.61) [15]. In community dwelling men age older than 70, those with both storage and voiding symptoms had a higher rate of falls than those without [16]. A systematic review of the association between LUTS and falls, injuries, and fractures in men concluded that both UI and LUTS are associated with falls in older men, with evidence that urgency, nocturia and frequency were consistently associated with falls, but only frequency was associated with fractures [17].

Despite this well-described association, potential underlying causes remain unexplained and poorly explored [18]. It has been proposed that people with LUTS may rush to the toilet and trip [19, 20], or falls occur after being incontinent and then slipping in the resultant pool of urine [21]. However, available evidence suggests that the majority of falls in older adults with LUTS do not occur during toileting [22], and a Japanese study examining falls in people with Parkinson's Disease and LUTS found that only 14% of falls occurred when getting to a toilet [23]. The belief that urinary urgency leads to rushing and therefore falls is also held by patients, with respondents in a qualitative study reporting that "having to rush for the toilet . . . because of a weak bladder . . . is a cause for falls" [24].

Neither continence nor walking are completely automatic processes. Maintaining continence relies on processing sensory input from the urothelium and detrusor, in multiple areas of the brain including the periaqueductal grey matter, the frontal and prefrontal cortices, and the pons [25]. Despite being largely automatic [26], gait and balance require highly complex integration of sensory information from the vestibular, ocular, and proprioceptive systems, all integrated in the frontal and parietal regions of the brain [27]. Incontinence and falls have been shown to be more common in those who are frail [28, 29], cognitively impaired [30, 31], and in those with cerebral white matter disease [32, 33].

As such, both maintaining continence and walking without falling are tasks which require active sensory input and cognitive processing. When two cognitive tasks are performed simultaneously, the speed or quality of performance of one or both tasks is reduced; a

concept known as dual-tasking or divided attention [34]. Dual tasking is associated with changes in gait and increased falls risk in older people. These changes in gait, such as reduced speed and stride length, and changes associated with an increased falls risk such as increased forward lean, have been demonstrated when secondary cognitive tasks are combined with walking [35–38]. The impact of divided attention on gait is greater in older than in younger adults.

When experiencing a strong desire to void (SDV), continent middle-aged women will slow their gait, not accelerate, and their step length decreases with an increase in gait variability compared to when walking with an empty bladder [39]. A study of continent and incontinent women aged 65 years and over found that SDV influenced gait parameters in both groups, with shorter stride length and increased stance time when experiencing SDV, and that the incontinent group had a slower self-selected gait speed at baseline [40].

We hypothesised that the sensation of urinary urgency acts as a source of divided attention in older adults with overactive bladder (OAB), that urgency will cause similar gait changes to distraction and that this, at least in part, explains the observed association between falls and LUTS in older people.

## Methods

### Recruitment and ethics

Participants were recruited from The Glenrose Continence Clinic in Edmonton, Alberta, Canada—a specialist geriatrics continence clinic, by advertising in the local press, and by approaching local seniors' associations. Men and women were included if they were aged 65 or over, and had OAB-wet, defined as per the International Continence Society (ICS) definition, with a daytime micturition frequency of eight or more, and urgency incontinence of at least once per week. Exclusion criteria were cognitive impairment, defined as a Montreal Cognitive Assessment Score (MoCA) of less than 26, executive dysfunction, defined as more than one error in the executive function parts of the MoCA (backward digit span, trail-marking test, word similarities, and word list generation), pharmacological treatment of OAB with either anticholinergic or beta-3-adrenergic medication, the inability to walk 30 metres independently and without aids, the use of a urinary catheter, or dialysis with anuria, a diagnosis of neurological disease that may affect gait, such as Parkinson's, previous stroke or multiple sclerosis, or sensory impairment such as visual or hearing loss sufficient to interfere with the conduct of the study. Recruitment and data collection took place between February 2018 and February 2019.

The study was approved by the University of Alberta Heath Research Ethics Committee, reference number PRO00054370.

### Sample size calculation

Verghese *et al*. [41] studied gait velocity under dual-task conditions in older adults and found a reduction from a mean of 104.7cms$^{-1}$ (SD 17.42) while walking with no distraction, to 72.2cms-$^1$ (SD 28.17) while walking under dual-task conditions. The Cohen's d for effect size of these data is 1.388. Assuming a similar change in our study population, with $\alpha = 0.5$ and $\beta = 0.8$, and using participants as their own controls, our minimum required sample size was 10, calculated using G*Power [42].

Given that the effect size associated with urinary urgency in the older adult population is unknown, we sought to over-recruit participants by 100% in order to minimise our risk of type 2 error.

## Instrumentation

Gait was assessed using 3Dimensional Instrumented Gait Analysis (3D IGA). This technique uses multiple cameras to record the positions of small reflective markers attached to the skin or clothing over bony prominences on the body [43]. Based on the findings of a pilot of this experimental method in older women experiencing urgency, markers were placed on the feet between the 2nd and 3rd metatarsal heads and posterior aspect of the calcaneus bilaterally, and on C7 posteriorly and the sternal notch anteriorly to measure movement of the trunk. Computer software (Visual 3D Professional, C-Motion, Inc., Germantown, MD, USA) allows for processing of these data and produces a three-dimensional image of the person's motion. From this, highly accurate measurements of the position of the body in time and space can be taken and converted to temporal-spatial and kinematic measurements [44]. The optical systems were calibrated at the beginning of each session of data collection and the markers were placed by an experienced and expert gait analyst (JL) in accordance with the gait laboratory's standard procedures.

Participants walked the length gait lab (9.1m, 30ft) thrice wearing their normal footwear and the final three gait cycles (six steps) were analysed. This was to allow the participants to get to a steady, self-selected speed, and to ensure sufficient gait cycles were available in case of camera or other data capture failure. The participants were not told which cycle was used for analysis to maintain data integrity. Participants were asked to walk at a comfortable pace until they had passed a mark on the gait lab floor, to reduce the possibility of participants slowing or changing gait as they approached the mark.

## Temporal-spatial gait data

For each step, the point of heel strike was identified and the position of the midfoot at this time marked. From this, velocity, cadence, stride length, and step width were calculated for each step and the mean value of the three gait cycles recorded.

## Kinematic data

Trunk lean was quantified by measuring the angle subtended the C7 and sternal notch markers, with the mean and range recorded. Foot-floor angle, defined as the angle subtended by the calcaneal and metatarsal marker at the point of heel strike, was also recorded. These were selected *a priori* to assess front-to-back lean, a measure associated with increased falls risk, and as a measure of a "flat-footed" or shuffling gait. A pilot study by our group found that more detailed kinematic data of leg motion was highly variable between individuals and did not provide useful data for analysis [45].

## Design

This was a within-subject repeated measure design. Temporal-spatial and kinematic gait parameters were recorded using 3D IGA, during walking under three conditions, *undistracted*, *distracted*, and when experiencing urinary *urgency*.

## Study procedure

Following informed, written consent, participants underwent multichannel subtracted cystometry according to standard ICS-approved protocol [46]. This was interpreted by an independent clinician and categorised according to the presence or absence of detrusor overactivity (DO) to allow a subgroup analysis and investigation of the influence of urgency due to DO on any gait changes. Those who declined to have this test were not excluded from gait analysis.

Participants then attended the gait laboratory, where a research assistant completed the Berg Balance Score (BBS), the Activities-specific Balance Confidence (ABC) score, and a LUTS severity score, the sex-specific International Consultation on Incontinence Questionnaire (ICIQ), and were asked about any falls, trips, or stumbles in the previous three months. The storage and incontinence subscales and associated bother scores were extracted from the sex-specific ICIQ questionnaires.

Reflective markers were applied and the participant then underwent gait analysis under three conditions; undistracted and with an empty bladder, when being distracted by the auditory n back test, and when experiencing urinary urgency. To reduce ordering effects, the order of state was determined by the blind drawing of lots at random. The study procedure was explained again and the location of the nearby toilet facilities shown. Participants were asked to empty their bladder immediately before data collection for the bladder empty and distracted walks.

The auditory n back test is a validated source of divided attention [47]. To perform the auditory n back test the examiner reads a list of letters aloud at a comfortable volume. When a letter which is the same as the letter 2 prior in the sequence is read out, the participant indicates verbally that this has occurred. So, in the sequence "F, B, D, E, D, A, C. . ." the second D would elicit a positive response. The n-back test relies on working memory and attention, and is a validated source of distraction; for the purpose of gait analysis, attempting the n-back test itself induces distraction irrespective of the participant's performance on the test [48]. Participants were asked to concentrate on the n-back test and to indicate as accurately as they could when a letter was repeated in the correct position. They commenced walking after the research assistant began reading the list of letters for the n back test and the n back continued until the participant had completed walking the gait lab.

To induce urgency, participants drank non-caffeinated fluids *ad libitum* at a comfortable pace until they experienced a compelling desire to void that was difficult to defer. When participants indicated that they needed to void, the examiner checked that they were unable to delay voiding any longer. If the participant did not absolutely have to go to void, they were encouraged to wait until they did. At this point they undertook gait analysis, with the final walk being towards the toilet. All three walks were completed on the same visit to the gait laboratory.

### Statistical analysis

For each gait parameter, the mean velocity, cadence, stride length, and step width were compared from the bladder empty condition to urgency and to distraction using two-tailed paired samples t-tests, having demonstrated normality with the Shapiro-Wilk test. Statistical significance was pre-defined at $p < 0.05$. The primary outcome measure was the change in gait velocity under each of the three conditions, as this has been shown to be influenced by urinary urge [39, 40]. Trunk lean data were not normally distributed and were compared using two-tailed Wilcoxon's signed rank tests. Subgroup analysis by the presence or absence of DO was performed using a mixed ANOVA for velocity, the primary outcome measure, using the classification of DO/Non-DO as a between-subject factor, to allow the impact of DO as a proxy for true urgency rather than urge to be assessed. Analysis was performed with using SPSS v25 (IBM Corp, Armonk, NY).

### Results

27 participants, 22 female and 5 male, were recruited and all successfully completed data collection. Their mean age was 75 years (SD 5.9). 7 participants had evidence of detrusor overactivity (DO), and in 14 DO was absent. 6 participants (3 men, 3 women) declined to undergo

multichannel cystometry. In the female participants, the mean ICIQ F-LUTS storage symptom score was 7.8 (SD 2.2) and mean bother score 20 (SD 7.3), and the incontinence score 9.2 (SD 4.3) and bother 27.9 (SD 13.2). In the male participants, the equivalent mean scores were 7.2 (SD 1.2), 25.6 (SD 6.0), 7.0 (SD 2.1) and 24.2 (SD 4.9) respectively. The mean Berg Balance Score was 52.5 (SD 3.0) suggesting functional balance [49] and Activities-specific Balance Confidence Score 83.2% (SD 16.4). No participants reported any slips, trips, or falls in the three months prior to recruitment. These results are summarised in Table 1.

### Adverse events

There were no trips or falls during gait analysis and no episodes of incontinence. No participant developed a symptomatic urinary tract infection following cystometry. All participants tolerated fluid loading well without episodes of nausea or vomiting.

### Gait analysis

**Temporal-spatial data.** Self selected gait velocity decreased from $1.1ms^{-1}$ at baseline to $1.0ms^{-1}$ when experiencing urgency, and $0.8 ms^{-1}$ when distracted. The change from baseline to both states was statistically significant (p = 0.008 and p<0.001). Likewise, stride length decreased, from 1.19m to 1.12m with urgency and 1.0m with distraction (p<0.001 for both comparisons). Cadence was significantly reduced by distraction (110steps/min to 94 steps/min, p<0.001) but not by urgency. Step width was unaffected by urgency (10.8cm to 10.9cm, NS) but was increased by distraction, to 12.0cm (p = 0.25). The presence of DO as a between-subject factor was non-significant (p = 0.77). These results are summarised in Table 2.

**Kinematic data.** One participant was too tall for kinematic data collection of trunk lean, as the C7 marker left the field of view of the camera during walking. Neither urgency nor distraction increased the range of lean, indicating no increase in front-to-back sway while walking. These data are summarised in Table 3.

**Table 1. Demographic data.**

| n = 27, 22 female, 5 male | | | |
|---|---|---|---|
| | **Mean** | **SD** | **Range** |
| Age | 75 | 5.9 | 65–87 |
| Montreal Cognitive Assessment Score | 27.6 | 1.4 | 26–30 |
| Berg Balance Score | 52.5 | 3.0 | 46–56 |
| Activities-specific Balance Confidence Score | 83.2 | 16.4 | 46–99 |
| ICIQ F-LUTS Storage Score | 6.76 | 2.21 | 2–11 |
| ICIQ F-LUTS Storage Score Bother | 20.1 | 7.27 | 3–28 |
| ICIQ F-LUTS Incontinence Score | 9.2 | 4.32 | 3–18 |
| ICIQ F-LUTS Incontinence Score Bother | 27.9 | 13.2 | 4–50 |
| ICIQ M-LUTS Storage Score | 7.2 | 1.3 | 6–9 |
| ICIQ M-LUTS Storage Score Bother | 25.6 | 6.07 | 17–33 |
| ICIQ M-LUTS Incontinence Score | 7 | 2.12 | 4–9 |
| ICIQ M-LUTS Incontinence Score Bother | 24.2 | 4.87 | 19–31 |

ICIQ F-LUTS/M-LUTS: International Consultation on Incontinence Questionnaire Lower Urinary Tract Symptoms Female/Male.

**Table 2. Temporal spatial gait analysis.**

| Gait Parameters (n = 27) | Baseline (mean (SD)) 95% CI | Urgency (mean (SD)) 95% CI | Distraction (mean (SD)) 95% CI | Baseline to Urgency Significance Mean Difference (95% CI) | Baseline to Distraction Significance Mean Difference (95% CI) | Urgency to Distraction Significance Mean Difference (95% CI) |
|---|---|---|---|---|---|---|
| Velocity (m/s) | 1.1 (0.16) | 1.0 (0.15) | 0.8 (0.19) | p = 0.008 | p<0.001 | p<0.001 |
|  | 1.02–1.15 | 0.96–1.07 | 0.72–0.87 | 0.9 (0.02–0.13) | 0.29 (0.19–0.4) | 0.2 (0.15–0.174) |
| Cadence (steps/min) | 110 (9.08) | 108 (11.2) | 94 (18.14) | p = 0.805 | p<0.001 | p<0.001 |
|  | 106–113 | 104–113 | 87–101 | 1.3 (-1.7–4.3) | 16 (8.0–23.9) | 1.3 (-4.3–1.6) |
| Stride Length (m) | 1.19 (0.16) | 1.12 (0.13) | 1.0 (0.13) | p<0.001 | p<0.001 | p<0.001 |
|  | 1.12–1.24 | 1.07–1.17 | 0.96–1.05 | 0.65 (0.03–0.99) | 0.17 (0.13–0.23) | 0.12 (0.73–0.16) |
| Step Width (cm) | 10.8 (0.7) | 10.9 (0.7) | 12.0 (0.6) | p>0.99 | p = 0.25 | p = 0.154 |
|  | 9–12 | 9–12 | 11–13 | 0 (-0.9–1.2) | 1.2 (0.1–2.3) | 1.1 (-3–2.4) |

Repeated-measures ANOVA with Bonferroni correction.

**Table 3. Kinematic gait analysis.**

| Kinematic Measure (n = 26) | | | | Significance | | |
|---|---|---|---|---|---|---|
|  | Baseline (mean (SD)) | Urgency (mean (SD)) | Distraction (mean (SD)) | Baseline to Urgency | Baseline to Distraction | Urgency to Distraction |
| Foot Floor Angle (°) | 22 (4.5) | 21.5 (3.68) | 19.0 (4.22) | p = 0.044 | p<0.001 | p<0.001 |
| C7-Sternal Angle mean (º) | 34.7 (7.0) | 35.8 (7.3) | 35.3 (7.7) | p = 0.012 | p = 0.23 | p = 0.23 |
| C7-Sternal Angle range (º) | 6.0 (2.4) | 6.1 (1.6) | 6.5 (2.6) | p = 0.82 | p = 0.44 | p = 0.92 |

Wilcoxon Signed-Rank Tests for pairwise comparison with Bonferroni correction.

## Discussion

These results demonstrate that gait velocity and stride length are similarly affected by both urgency and divided attention in older adults with OAB. It is therefore likely that the sensation of urgency acts as a source of divided attention in older adults with OAB, with those experiencing urgency devoting cognitive resource to maintaining continence at the cost of deteriorating gait.

Participants had moderate LUTS, based on the mean filling subscale score of 7.8/15, with moderate degree of bother. Participants' BBS and ACB scores indicated that that they were at low risk of falls and were highly confident in their balance [50].

In this sample of older adults with OAB, we demonstrated a *decrease* in velocity and step length with both urgency and divided attention, changes which are associated with an increase in falls risk [51]. This is similar to the effect observed with SDV in continent, middle-aged women [39], and these findings add further evidence disputing the notion that urinary urgency and urgency incontinence may lead to falls by inducing people to rush or run to the toilet [20]. The observed small decrease in foot-floor angle (FFA) with both urgency and distraction is unlikely to be clinically significant; although no normative data for FFA exist in adults, the reported standard deviation of FFA in children is 2.8˚ [52]. The observed C7-Sternal angle changes indicated that people leant forwards when walking with urgency. Contracting the pelvic floor induces a posterior pelvic tilt, and it is possible that our participants

compensated for that by leaning their trunk forward, although we were unable to assess pelvic floor contraction directly.

Divided attention induces deleterious gait changes in older people [53]. How divided attention causes decline in simultaneous tasks is debated in the literature. Briefly, there are three main models; *capacity sharing*, which suggests the brain has a finite capacity for global function, and if simultaneous tasks exceed this threshold, performance declines, *bottleneck (or task-switching)*, which suggests that individual brain areas can only perform one function at a time, so if the competing tasks require the same pathway, a bottleneck occurs, slowing processing, and *cross-talk model*, which suggests that simultaneous tasks are more difficult if they both require similar sensory input [34]. In all these models, prioritisation occurs, in that a subconscious decision is made to devote greater cognitive resource to one task over another. In this study, none of the participants experienced incontinence, suggesting that our participants prioritised their bladder control over gait, hence the deterioration in gait parameters.

It is always challenging to differentiate between *urgency*, the sudden compelling desire to void that is difficult to defer, and *urge*, the physiological sensation of a full bladder. All our participants had a diagnosis of OAB with urgency and urgency incontinence, but we made no attempt to quantify the desire to void with a visual analogue scale (VAS) or similar tool such as the Urgency Sensation Scale, as we explained clearly that the participant should delay voiding until they could absolutely not delay further, which would by definition correspond to a 4 on the USS or a 10 on a VAS, and that adding a further test between urgency and walking may have increased the risk of urgency UI during the study.

We used multi-channel pressure subtracted cystometry to classify our participants into DO/non-DO, as the finding of DO would increase our confidence that pathological urgency was being reported. However, the presence or absence of DO on cystometry had no effect on the results, suggesting that either our participants were experiencing true urgency, or that any SDV in older people with OAB acts as a source of divided attention, whether it be a strong sensation of urinary urge or pathological urgency.

Despite calls, no intervention trial has been performed to investigate whether treating OAB reduces the risk of falls. Possible interventions include pharmacological management of OAB or a potential for dual task training, which has been shown to improve gait under conditions of divided attention [54]. This may not only reduce the risk of falls but also allow control of OAB symptoms. Dual task training has been shown to improve executive function testing and dual-task gait in women with mixed urinary incontinence [55].

## Strengths and limitations

This is the first study to use 3D-IGA to record temporal-spatial and kinematic gait data from older people with OAB and compare gait under the conditions of urgency and distraction. 3D capture technology allows highly accurate measurements of gait and joint position, and therefore gait velocity. We averaged the final 3 gait cycles of the walk for analysis and were therefore unable to assess gait variability, which has been previously shown to increase with urinary urge [39] and divided attention [56]. The study is limited by its small sample size, although we over-recruited based on the sample size calculation, the small number of participants with demonstrable DO on cystometry limits the confidence in the conclusions from the subgroup analysis by presence or absence of DO. Although all our participants had a diagnosis of OAB and were asked to wait until they were experiencing a compelling desire to void that was difficult to defer, it is not possible to differentiate between true urgency and a strong desire to void or urge in an experimental model.

Although we met the target for recruitment from our sample size analysis, we included both men and women in the study, and our sample was biased towards female participants. Although all participants had a diagnosis of OAB, it is recognised that there are sex differences in LUTS, with women experiencing more stress and mixed incontinence than men [10]. There were insufficient numbers recruited to allow subgroup analysis by sex. Future studies may wish to recruit sufficient participants to facilitate such sex-based analysis.

None of the participants reported having experienced falls, and their Berg Balance Score and Activities-specific Confidence Scores were consistent with low falls risk. As such, it is difficult to directly associate the gait changes observed in this population with a significant increase in absolute falls risk. Repeating this study in older people with OAB who have experienced falls would be useful.

## Conclusion

These results demonstrate that, in older adults with OAB, the sensation of urinary urgency induced similar changes in gait to a known source of distraction, suggesting that urgency acts as a source of divided attention in this group. It is well established that divided attention increases the risk of falls in older people, and therefore urgency acting as a distractor may, in part, explain the known association between falls and LUTS in older people.

## Acknowledgments

The authors would like to thank Mr Justin Lewicke for his expertise in motion capture and gait analysis.

## Author Contributions

**Conceptualization:** William Gibson, Kathleen Hunter, Adrian Wagg.

**Data curation:** William Gibson.

**Formal analysis:** William Gibson, Allyson Jones, Kathleen Hunter, Adrian Wagg.

**Funding acquisition:** William Gibson, Kathleen Hunter, Adrian Wagg.

**Investigation:** William Gibson, Adrian Wagg.

**Methodology:** William Gibson, Kathleen Hunter.

**Project administration:** William Gibson.

**Resources:** Adrian Wagg.

**Supervision:** Allyson Jones, Kathleen Hunter, Adrian Wagg.

**Writing – original draft:** William Gibson.

**Writing – review & editing:** William Gibson, Allyson Jones, Kathleen Hunter, Adrian Wagg.

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
