## [Decision Letter · Decision Letter 0]

11 Jun 2021

PONE-D-21-12403

Urinary urgency acts as a source of diverted attention leading to changes in gait in older adults with overactive bladder

PLOS ONE

Dear Dr. Gibson,

Thank you for submitting your manuscript to PLOS ONE. After careful consideration, we feel that it has merit but does not fully meet PLOS ONE’s publication criteria as it currently stands. Therefore, we invite you to submit a revised version of the manuscript that addresses the points raised during the review process.  

We look forward to receiving your revised manuscript.

Kind regards,

Jean L. McCrory, PhD

Academic Editor

PLOS ONE

Journal Requirements:

2. Thank you for stating in your manuscript text that informed consent was obtained. In the ethics statement in the Methods and online submission information, please ensure that you have specified what type of consent you obtained (for instance, written or verbal, and if verbal, how it was documented and witnessed).

3. In your Methods section, please provide additional information about the participant recruitment method and the demographic details of your participants. Please ensure you have provided sufficient details to replicate the analyses such as:

a) the recruitment date range (month and year),

b) the name of the specialist continence clinic where participants were recruited from

Reviewers' comments:

Reviewer's Responses to Questions

**Comments to the Author**

1. Is the manuscript technically sound, and do the data support the conclusions?

Reviewer #1: Yes

Reviewer #2: Yes

2. Has the statistical analysis been performed appropriately and rigorously? 

Reviewer #1: Yes

Reviewer #2: Yes

3. Have the authors made all data underlying the findings in their manuscript fully available?

Reviewer #1: Yes

Reviewer #2: Yes

4. Is the manuscript presented in an intelligible fashion and written in standard English?

Reviewer #1: Yes

Reviewer #2: Yes

5. Review Comments to the Author

Reviewer #1: Urinary urgency acts as a source of diverted attention leading to changes in gait in older adults with overactive bladder

Summary: This study explores differences between walking, walking with a cognitive task and walking with OAB. Findings demonstrate that walking with OAB is similar to walking under divided attention conditions with a cognitive (n-back) task. As there are few studies that examine walking with urinary incontinence and its implication for falls, this work is critical to share with the research community. Certain components of the manuscript need clarification for publication in PloS One but it is this reviewers’ opinion that this work should be shared with the research community once these clarifications are made.

Abstract

- Conclusion: see comment in conclusion section below.

Introduction

- The second paragraph should be divided into more than one sentence. After references [8,9], the authors should specify the age group and then the next sentence should be about the prevalence etc.

- 4th paragraph: I think this sentence lacks context in terms of the relationship between falling, Parkinson’s and LUTS.

“In people with Parkinson’s Disease and LUTS only 14% of falls occurred when getting to a toilet [20].”

-4th paragraph: there is some qualitative evidence that older women report having a weak bladder is a cause for falls (see page 409: Muhaidat J, Skelton D, Kerr A, Evans J, Ballinger C (2010) Older adults’ experiences and perceptions of dual tasking. British Journal of Occupational Therapy, 73(9), 405-412.

- 5th paragraph: Is this compared to a group of incontinent middle-aged women?

“When experiencing a strong desire to void (SDV), continent middle-aged women will slow their gait, not accelerate, and their step length decreases with an increase in gait variability [21].”

- Dual task is more commonly referred to as divided attention rather than diverted attention. See: Fraser SA, Bherer L. Age-related decline in divided attention: From theoretical lab research to practical real life situations. An advanced review of divided attention. In: Nadel L, ed. Wiley Interdisciplinary Reviews: Cognitive Science. New York: Wiley Interscience; 2013; 4:623–640. I think it is in the authors’ best interests to use the term divided attention rather than diverted throughout the manuscript, as this may attract other readers that may know less about urinary incontinence but be interested from a dual-task gait in older adults’ perspective. I personally think that consideration for urinary incontinence as factor dividing the individual’s attention is something important to consider for gait and dual-task gait.

- 7th paragraph: I think a stronger link could be made between dual tasking and UI. Rather than comparing younger and older adults, what characteristics of UI and falling are similar to dual tasking and falling? I think the next paragraph makes this clearer.

- For people who are unfamiliar with the topic, I think the differences between urinary incontinence (UI), lower urinary tract symptoms (LUTS) and overactive bladder (OAB) are unclear.

- Hypothesis could be more specific. Was the expectation that walking with OAB would be similar to walking with a cognitive task? In which case both conditions would be considered divided attention conditions? As with comments above, I think stating this explicitly will help the reader understand the link between divided attention and falls – and between dual task gait with a cognitive task or OAB being similar and potentially increasing falls risk.

Methods

- I think the “Design” section would fit better in the “Study Procedure” section so that the authors can describe the three conditions immediately after mentioning them.

- Recruitment, “daytime frequency of 8 or more” – 8 or more what? Occasions of urinary urgency? Leaks?

- 2nd paragraph of the “instrumentation” section: Could analyzing the last 3 gait cycles affect gait measures given that participants are likely decelerating as they reach the end of the lab?

- While I agree with this statement: “From this, highly accurate measurements of the position of the body in time and space can be taken and converted to temporal-spatial and kinematic measures.” It would be nice to have a citation about the validity of this technique.

- 4th paragraph of the “study procedure” section: In the dual-task literature, it is important whether the responses are correct because correct responses could mean that the task wasn’t distracting enough or hypothetically induce a fall in the case of UI. Similarly, consistently incorrect responses could mean that participants gave up on the n back task (or bladder control) in favour of walking. Perhaps the authors want to frame the experiment as having a focus on gait being the primary task and the cognitive task secondary?

- Also in study procedure, to be clear, in the condition where the participants’ attention is divided between the n-back task and walking – the participants also have an empty bladder?

- Can the authors clarify what type of instruction, if any, was given to participants when walking? Were they instructed to walk at a self-selected pace? Were they instructed to respond as quickly and as accurately as possible to the n-back? Please clarify instructions. Instructions are very important for prioritization of different tasks during divided attention.

Statistical analysis

- Unless this is a requirement of the journal, I think the “sample size calculation” section belongs in the “recruitment and ethics section” of the methods.

Results

- For the ICIQ F-LUTS, I know the sample is imbalanced with more women than men, but I wonder if the values for this scale that were found are expected (this may be a discussion point) – Are there sex differences in this score? Does the data replicate what is already in the literature?

Discussion

- First sentence might be missing the word “results” or “findings” after “these”?

- 3rd paragraph: I’m having trouble following the comparison between middle-aged, continent women with SDV and the potential of falling in the participants with LUTS. I think this could also be explained more thoroughly in the fourth and fifth paragraphs of the introduction, and discussion and conclusion.

“We demonstrated a decrease in velocity and step length with both urgency and diverted attention, which is similar to the effect observed with SDV in continent, middle-aged women [21]. These findings add further evidence to dispute the notion that urinary urgency and urgency incontinence may lead to falls by inducing people to run to the toilet [17].”

The introduction and discussion imply that rushing to the toilet (i.e., gait velocity) is not the cause of falls in older adults with LUTS and is similar to middle-aged women with SDV. How can diverted attention then lead to falls if the costs of diverted attention (i.e., slower gait velocity and step length), don’t lead to falls? The next paragraph discusses prioritization. Does prioritizing bladder over gait lead to falls? Is there a difference between middle-aged participants and the older adults from this study?

- Interesting that in the discussion the authors discuss prioritization – which in the case of this study is difficult to decipher since the cognitive task performance was not measured and the instructions to participants during divided attention (n-back) and urgency are not clear.

- Although not specific to OAB, I would add that dual task training has improved mixed-UI in women with particular effects to dual-task gait – and that this shows promise for OAB…Reference: Fraser, S. A., Elliott, V., de Bruin, E. D., Bherer, L., & Dumoulin, C. (2014). The effects of combining videogame dancing and pelvic floor training to improve dual-task gait and cognition in women with mixed-urinary incontinence. Games for Health: Research, Development, and Clinical Applications, 3(3), 172-178.

Conclusion

- I think the conclusion is incomplete. The association between falls and LUTS is unclear based on the discussion and measures of gait that are similar to continent people.

- Also none of the sample reported falls – so it is difficult to make this conclusion. Perhaps in limitations, state that none had reported previous falls – limiting the connection between these findings and falls. Then argue that follow-up longitudinal studies would help to clarify if the divided attention findings reported here (particularly those with OAB during walking) lead to an increased falls risk.

Reviewer #2: A well written and innovative paper providing new insights and novel data towards explaining the mechanism underpinning the relationship between falls and overactive bladder in older people.

A few minor suggestions are made:

Please clarify the funding source in the paper - PlOS ONE financial disclosure is different to the paper.

For clarification throughout the paper suggest insert 'urinary' before each use of urgency, as results would not apply to 'faecal' urgency.

P 11 Instrumentation- please insert distance of gait lab walk eg 10 m?.

Table 1 - please insert explanation for DO to key

Table 2 - please insert explanation for SDV to key or change to 'urgency'

Table 3 - amend layout to avoid splitting words and numbers

P 13 - There does not seem to be any subgroup analysis by presence or absence of DO presented.

P 19 Discussion - suggest move 'in older adults with OAB' at end of second sentence to first sentence following '.... stride length.

P 19 discussion 3rd paragraph - insert 'In this sample of older adults with OAB' before 'we demonstrated a decrease....etc'

P 19 discussion 4th paragraph - last sentence - clarify what 'Both' refers to.

P 20 Discussion paragraph 6 - by definition urgency is a 'sudden' compelling desire to void etc whereas in this study participants were delaying voiding until they could hold no longer, and presumably this was a gradually developing sensation - perhaps 'urgency' as used might be better described as a strong desire to void?

P 20 paragraph 7 - there are no results presented on the impact of cystometric DO/non DO classification yet they are discussed.

6. PLOS authors have the option to publish the peer review history of their article (what does this mean?). If published, this will include your full peer review and any attached files.

Reviewer #1: No

Reviewer #2: No

---

## [Author Response · Author response to Decision Letter 0]

13 Jul 2021

We have addressed the reviewers’ comments as follows.

Journal Requirements

 Thank you for stating in your manuscript text that informed consent was obtained. In the ethics statement in the Methods and online submission information, please ensure that you have specified what type of consent you obtained (for instance, written or verbal, and if verbal, how it was documented and witnessed).

Participants gave written, informed consent. We have added this to the description of the consent.

 3. In your Methods section, please provide additional information about the participant recruitment method and the demographic details of your participants. Please ensure you have provided sufficient details to replicate the analyses such as:

a) the recruitment date range (month and year),

b) the name of the specialist continence clinic where participants were recruited from

The name of the clinic was omitted for anonymity; we have now added this. The dates of recruitment have also been added as requested. We have also added the name of the HREB.

Reviewer 1:

 - The second paragraph should be divided into more than one sentence. After references [8,9], the authors should specify the age group and then the next sentence should be about the prevalence etc.

We have separated the paragraph into two sentences, and added age-related prevalence of urgency for context.

 - 4th paragraph: I think this sentence lacks context in terms of the relationship between falling, Parkinson’s and LUTS.

This sentence has been rewritten to add context.

 -4th paragraph: there is some qualitative evidence that older women report having a weak bladder is a cause for falls (see page 409: Muhaidat J, Skelton D, Kerr A, Evans J, Ballinger C (2010) Older adults’ experiences and perceptions of dual tasking. British Journal of Occupational Therapy, 73(9), 405-412.

Thank you for pointing this out. The observation that patients also share the belief that rushing is a factor has been added.

 - 5th paragraph: Is this compared to a group of incontinent middle-aged women?

“When experiencing a strong desire to void (SDV), continent middle-aged women will slow their gait, not accelerate, and their step length decreases with an increase in gait variability [21].”

These changes are compared to the bladder-empty state in the same individual. This has been clarified.

- Dual task is more commonly referred to as divided attention rather than diverted attention. See: Fraser SA, Bherer L. Age-related decline in divided attention: From theoretical lab research to practical real life situations. An advanced review of divided attention. In: Nadel L, ed. Wiley Interdisciplinary Reviews: Cognitive Science. New York: Wiley Interscience; 2013; 4:623–640. I think it is in the authors’ best interests to use the term divided attention rather than diverted throughout the manuscript, as this may attract other readers that may know less about urinary incontinence but be interested from a dual-task gait in older adults’ perspective. I personally think that consideration for urinary incontinence as factor dividing the individual’s attention is something important to consider for gait and dual-task gait.

Thank you for this helpful insight. The term “diverted” has been replaced with “divided” throughout as suggested. We have also edited the title of the paper..

- 7th paragraph: I think a stronger link could be made between dual tasking and UI. Rather than comparing younger and older adults, what characteristics of UI and falling are similar to dual tasking and falling? I think the next paragraph makes this clearer.

- Hypothesis could be more specific. Was the expectation that walking with OAB would be similar to walking with a cognitive task? In which case both conditions would be considered divided attention conditions? As with comments above, I think stating this explicitly will help the reader understand the link between divided attention and falls – and between dual task gait with a cognitive task or OAB being similar and potentially increasing falls risk.

Thank you for this suggestion and observation. The cognitive aspects of continence were in the discussion. This paragraph has been moved to the introduction to strengthen and clarify the reasons for the hypothesis, and the hyporthsis has been further clarified.

- For people who are unfamiliar with the topic, I think the differences between urinary incontinence (UI), lower urinary tract symptoms (LUTS) and overactive bladder (OAB) are unclear.

Definitions of these terms have been added at appropriate points.

- I think the “Design” section would fit better in the “Study Procedure” section so that the authors can describe the three conditions immediately after mentioning them.

This has been moved as suggested.

- Recruitment, “daytime frequency of 8 or more” – 8 or more what? Occasions of urinary urgency? Leaks?

Frequency has a specific meaning in continence literature. This has been clarified.

- 2nd paragraph of the “instrumentation” section: Could analyzing the last 3 gait cycles affect gait measures given that participants are likely decelerating as they reach the end of the lab?

This is possible. However, participants were asked to walk at a steady speed, and there was space past the end point to walk in to, rather than a hard stop. As participants were internal controls the effect of any end-of-walk deceleration would have been consistent across states and not affected the results. We have added to the description of the data collection to reflect this.

- While I agree with this statement: “From this, highly accurate measurements of the position of the body in time and space can be taken and converted to temporal-spatial and kinematic measures.” It would be nice to have a citation about the validity of this technique.

A reference has been added for this statement.

- 4th paragraph of the “study procedure” section: In the dual-task literature, it is important whether the responses are correct because correct responses could mean that the task wasn’t distracting enough or hypothetically induce a fall in the case of UI. Similarly, consistently incorrect responses could mean that participants gave up on the n back task (or bladder control) in favour of walking. Perhaps the authors want to frame the experiment as having a focus on gait being the primary task and the cognitive task secondary?

We believe that it is clear that the outcomes of this study are related to gait changes rather than cognitive performance, and the reference given in the text supports the sentence as written. We have clarified this in the text.

- Also in study procedure, to be clear, in the condition where the participants’ attention is divided between the n-back task and walking – the participants also have an empty bladder?

This is correct; we have added a sentence to clarify this.

- Can the authors clarify what type of instruction, if any, was given to participants when walking? Were they instructed to walk at a self-selected pace? Were they instructed to respond as quickly and as accurately as possible to the n-back? Please clarify instructions. Instructions are very important for prioritization of different tasks during divided attention.

The instructions given to participants have been clarified as requested.

Statistical analysis

- Unless this is a requirement of the journal, I think the “sample size calculation” section belongs in the “recruitment and ethics section” of the methods.

This has been moved as requested.

Results

- For the ICIQ F-LUTS, I know the sample is imbalanced with more women than men, but I wonder if the values for this scale that were found are expected (this may be a discussion point) – Are there sex differences in this score? Does the data replicate what is already in the literature?

The distribution of responses on the ICIQ-fLUTS and mLUTS will vary widely with sample population. They merely describe the distribution of LUTS within our sample. Anecdotally, they are broadly representative of the population attending our clinic, although we have not analysed these data formally. We recruited a mixed-sex sample, all with a diagnosis of OAB. There are sex-based differences in LUTS in the general population, with women experiencing more stress and mixed-incontinence. This has been added to the limitations section.

Discussion

- First sentence might be missing the word “results” or “findings” after “these”?

Thank you for noticing this unfortunate omission. The word “results” has been added.

- 3rd paragraph: I’m having trouble following the comparison between middle-aged, continent women with SDV and the potential of falling in the participants with LUTS. I think this could also be explained more thoroughly in the fourth and fifth paragraphs of the introduction, and discussion and conclusion.

This refers to the gait changes induced by urgency and distraction in this study, which are similar to those induced by SDV in another patient population in the work by Booth

-The introduction and discussion imply that rushing to the toilet (i.e., gait velocity) is not the cause of falls in older adults with LUTS and is similar to middle-aged women with SDV. How can diverted attention then lead to falls if the costs of diverted attention (i.e., slower gait velocity and step length), don’t lead to falls?

The observed changes in gait are associated with increased falls risk, and this has been added. The underlying finding of the paper is that urgency has similar impact on gait as divided attention, and it is well established that divided attention increases risk of falls in older adults. We have clarified this message at appropriate points. 

- Although not specific to OAB, I would add that dual task training has improved mixed-UI in women with particular effects to dual-task gait – and that this shows promise for OAB…Reference: Fraser, S. A., Elliott, V., de Bruin, E. D., Bherer, L., & Dumoulin, C. (2014). The effects of combining videogame dancing and pelvic floor training to improve dual-task gait and cognition in women with mixed-urinary incontinence. Games for Health: Research, Development, and Clinical Applications, 3(3), 172-178.

This has been added as suggested.

- I think the conclusion is incomplete. The association between falls and LUTS is unclear based on the discussion and measures of gait that are similar to continent people.

The conclusion has been rewritten to clarify the underlying message of the paper.

- Also none of the sample reported falls – so it is difficult to make this conclusion. Perhaps in limitations, state that none had reported previous falls – limiting the connection between these findings and falls. Then argue that follow-up longitudinal studies would help to clarify if the divided attention findings reported here (particularly those with OAB during walking) lead to an increased falls risk.

A sentence to the effect has been added to the conclusion. Longitudinal studies would be valuable to demonstrate a temporal relationship of developing OAB prior to an observable increase in falls risk, they would not help address the hypothesis of this paper. Further work to establish if dual-task training ameliorates the effect of urgency on gait would be useful, and is mentioned in the discussion.

Reviewer 2

Reviewer #2: A well written and innovative paper providing new insights and novel data towards explaining the mechanism underpinning the relationship between falls and overactive bladder in older people.

A few minor suggestions are made:

Thank you for this comment.

Please clarify the funding source in the paper - PlOS ONE financial disclosure is different to the paper.

The source of funding is given on the title page and has been corrected in the PLOS system.

For clarification throughout the paper suggest insert 'urinary' before each use of urgency, as results would not apply to 'faecal' urgency.

We have clarified this where appropriate. Whether or not faecal urgency has a similar distracting effect is the subject for another paper!

P 11 Instrumentation- please insert distance of gait lab walk eg 10 m?

The length of the gait lab has been added as requested.

Table 1 - please insert explanation for DO to key

The DO data has been removed from table 1 and clarified in the results section for clarity.

Table 2 - please insert explanation for SDV to key or change to 'urgency'

This has been changed as requested.

Table 3 - amend layout to avoid splitting words and numbers

P 13 - There does not seem to be any subgroup analysis by presence or absence of DO presented.

DO was used as a between-subject factor within the ANOVA as described in the methods. This has been added to the results for clarity.

P 19 Discussion - suggest move 'in older adults with OAB' at end of second sentence to first sentence following '.... stride length.

This has been added as suggested.

P 19 discussion 3rd paragraph - insert 'In this sample of older adults with OAB' before 'we demonstrated a decrease....etc'

This has been added as suggested.

P 19 discussion 4th paragraph - last sentence - clarify what 'Both' refers to.

This part of the discussion has been moved to the introduction, and “both” has been replaced with “incontinence and falls” for clarity.

P 20 Discussion paragraph 6 - by definition urgency is a 'sudden' compelling desire to void etc whereas in this study participants were delaying voiding until they could hold no longer, and presumably this was a gradually developing sensation - perhaps 'urgency' as used might be better described as a strong desire to void?

This is an excellent point and a perennial difficulty in this type of research. As our participants all have a diagnosis of OAB and were instructed to wait for a compelling desire to void which they found difficult to defer, we believe that we recreated true urgency as closely as possible. We have added this to the limitations section.

P 20 paragraph 7 - there are no results presented on the impact of cystometric DO/non DO classification yet they are discussed.

These results have been added.

---

## [Decision Letter · Decision Letter 1]

3 Sep 2021

Urinary urgency acts as a source of diverted attention leading to changes in gait in older adults with overactive bladder

PONE-D-21-12403R1

Dear Dr. Gibson,

We’re pleased to inform you that your manuscript has been judged scientifically suitable for publication and will be formally accepted for publication once it meets all outstanding technical requirements.

Kind regards,

Jean L. McCrory, PhD

Academic Editor

PLOS ONE

Additional Editor Comments (optional):

Reviewers' comments:

Reviewer's Responses to Questions

**Comments to the Author**

1. If the authors have adequately addressed your comments raised in a previous round of review and you feel that this manuscript is now acceptable for publication, you may indicate that here to bypass the “Comments to the Author” section, enter your conflict of interest statement in the “Confidential to Editor” section, and submit your "Accept" recommendation.

Reviewer #2: All comments have been addressed

2. Is the manuscript technically sound, and do the data support the conclusions?

Reviewer #2: Yes

3. Has the statistical analysis been performed appropriately and rigorously? 

Reviewer #2: Yes

4. Have the authors made all data underlying the findings in their manuscript fully available?

Reviewer #2: Yes

5. Is the manuscript presented in an intelligible fashion and written in standard English?

Reviewer #2: Yes

6. Review Comments to the Author

Reviewer #2: (No Response)

7. PLOS authors have the option to publish the peer review history of their article (what does this mean?). If published, this will include your full peer review and any attached files.

Reviewer #2: No

---

## [Editor Report · Acceptance letter]

24 Sep 2021

PONE-D-21-12403R1 

Urinary urgency acts as a source of divided attention leading to changes in gait in older adults with overactive bladder. 

Dear Dr. Gibson:

I'm pleased to inform you that your manuscript has been deemed suitable for publication in PLOS ONE. Congratulations! Your manuscript is now with our production department. 

Kind regards, 

on behalf of

Dr. Jean L. McCrory 

Academic Editor

PLOS ONE